## Perspective

dispersal limitation; ecological restoration; habitat loss; land-use legacies; reintroduction

**Corresponding author:**
Richard T. Corlett;
Email: rtcorlett@gmail.com

# Rewilding plants and vegetation

Richard T. Corlett[1,2]

[1]Center for Integrative Conservation and Yunnan Key Laboratory for the Conservation of Tropical Rainforests and Asian Elephants, Xishuangbanna Tropical Botanical Garden, Chinese Academy of Sciences, Yunnan, China and [2]Royal Botanic Gardens Kew, London, UK

## Abstract

Ecological restoration has traditionally had a bottom-up focus on plants and vegetation, but rewilding has been the opposite, and the impacts of rewilding carnivores and large herbivores on plant species and vegetation are largely unknown. The aim of this perspective, therefore, is to clarify what rewilding means for plants and vegetation, to assess progress in achieving this, to identify research needs and to make recommendations for rewilding practice. Land-use legacies and dispersal limitation are major challenges for plant rewilding, and the slowness of vegetation recovery makes success hard to evaluate on a human timescale. On the other hand, wild vegetation develops spontaneously wherever human pressures are released, regardless of the state of the site. For plant conservation, the key issue is ensuring that all plant species that can be restored are present, including rare and threatened species. Long-term species-level monitoring and, where necessary, continued intervention should be part of all projects that aim to rewild plants and vegetation.

## Impact statement

The rewilding literature focuses almost entirely on animals, with plants mentioned, if at all, as passive recipients of herbivory, trampling and seed dispersal services provided by animals. However, as increasing areas are subject to rewilding, it is important that the impacts on plant species and vegetation are understood, and that rewilding practices are modified, where necessary, in order to maximise benefits for the conservation of plants as well as animals. This perspective aims to encourage this by first clarifying what rewilding means for plants and vegetation, then assessing the extent to which this is being achieved with current practices, and finally identifying needs for changes to rewilding practices. Research needs are also addressed.

## Introduction

Ecological restoration has traditionally had a bottom-up focus on plants and vegetation, whereas rewilding has been the opposite (Nelson, 2024). If plants are mentioned, it is as passive recipients of herbivory, trampling and seed dispersal services provided by the focal animals. From its origin in the 'cores, corridors and carnivores' model of Soulé and Noss (1998) and more recent focus on megaherbivores, the emphasis in rewilding has been on top-down trophic effects (Svenning et al., 2024). The aim of this essay, therefore, is to bring a plant perspective to rewilding by first clarifying what rewilding means for plants and vegetation, then assessing progress in achieving this and finally identifying needs for research and rewilding practice.

## What is rewilding as currently practiced?

The aim of rewilding is to restore a self-sustaining ecosystem (Carver et al., 2021; Mutillod et al., 2024; Svenning et al., 2024). In contrast to classical ecological restoration, the focus is on recovery of ecological processes and interactions, particularly trophic interactions, rather than a particular species composition, so taxon substitutions may be made for functionally important species that are globally extinct (Mutillod et al., 2024). Human intervention is minimised, but it may be necessary initially to overcome the lasting effects of past human impacts and set the system on a desired trajectory. The subsequent management should be adaptive in response to evidence from monitoring. Continued or periodic interventions may be needed in smaller rewilded areas, or where practical constraints exist on re-introducing keystone herbivores or carnivores (Svenning et al., 2024). The aim is restoration of wildness to the maximum extent possible. The degree of rewilding, possible at a particular site, is envisaged on a linear scale from minimal to partial and near-full to full, depending on three main factors: the extent to which a natural trophic structure with competing herbivores and apex carnivores can be restored; the need for ongoing interventions and the degree to which the system becomes self-regulating (Pedersen et al., 2020).

## What is rewilding for plants and vegetation?

Plants differ from animals in many fundamental ways, but the most relevant difference is that plants are immobile, except at the seed dispersal stage. An animal can move away from the release site, whereas a plant cannot. In nature, plants typically depend on seed dispersal for initial establishment into suitable fine-scale micro-habitats, but most are also easily established artificially, even outside their natural range, and can then persist without recruitment for decades or longer. While this capacity is useful for agriculture and forestry, and for *ex situ* conservation in botanical gardens and arboreta, it can give a misleading impression of the success of restoration efforts. A skilled gardener can create a landscape that appears to be spontaneous and wild, but is not, as becomes obvious if maintenance is stopped.

In practice, there is a continuum between gardening, ecological restoration and rewilding. The growth of a sown seed or planted sapling is successful gardening and may contribute to ecological restoration, but it cannot be considered as rewilding until a self-sustaining population has been established. A fully rewilded plant population is one that no longer depends on human assistance. In practice, however, there will be degrees of rewilding, depending on the extent to which continued human support is required.

There have been proportionally far fewer post-Linnean extinctions for plants than for vertebrates (0.2% of plants compared with 5% of mammals and 7% of birds (Humphreys et al., 2019), so most plant species are still wild somewhere, but anthropogenic changes in the structure and species composition of vegetation are probably ubiquitous. In the modern world, 'natural' vegetation can be defined as vegetation that has not been deliberately planted or sown by people. This is a low bar, and above it extends a continuum with decreasingly obvious legacies of human impact, with the most natural vegetation impacted only by past herbivore extinctions, anthropogenic climate change and rising carbon dioxide. Natural vegetation by this definition also includes habitats that are often termed 'semi-natural': non-crop habitats modified by human use or management, including semi-natural grasslands grazed by livestock and forests managed for timber production.

Another major difference between plants and animals is that plants do things more slowly. Long lag times are inherent in plant and vegetation processes, making rewilding success difficult to assess (Albrecht et al., 2019). Multi-decadal life spans are much commoner in plants than in animals, and in woody plants, annual growth accumulates. Old trees, where they occur, have unique structural and functional roles and are irreplaceable on a centuries-long timescale (Schweiger and Svenning, 2020), but not all natural vegetation has trees, and not all natural forests have large old trees.

## Rewilding plant species

Planting is usually the fastest way to establish plants of a desired species and new vegetation of a desired composition, but these are not wild plants, and rewilding success cannot be judged until the second and subsequent generations. Although most plant reintroductions aim to create persistent, self-sustaining populations, evaluations of their success are usually based on short-term benchmarks, such as survival and reproduction of the founder generation (Bellis et al., 2024). The factors favouring initial establishment may differ from those needed for long-term persistence; thus, these benchmarks can be misleading. There are fast plants, including annuals, but for most species, the multi-generational monitoring needed to assess rewilding success will require decades or centuries.

On a human timescale, the rewilding of plant species is usually a work in progress.

Many plant reintroductions involve some degree of post-release aftercare; most often competition reduction, watering or grazer exclusion (Corli et al., 2023). As with supplementary feeding of vertebrates undergoing soft release, this is usually intended to be temporary. Long-term aftercare results in semi-wild populations: preferable to purely *ex situ* conservation because these populations can support associated species of animals and microbes, but not full rewilding.

## Rewilding vegetation

While plant reintroductions have focussed on rare species that, *a priori*, would be expected to be difficult to re-establish in the wild, most attempts to re-establish natural vegetation use dominant species and/or those for which planting material is most easily produced. Not only are these likely to be easier to establish, but the success of individual species becomes less important when multiple species are planted. Rewilding should therefore be easier for vegetation than for individual species. For herbaceous vegetation with relatively rapid turnover, it is possible to evaluate success within a decade or so, but few, if any, forest restoration projects have been running long enough for the planted trees to be replaced by wild plants.

Natural regeneration – 'passive rewilding' – establishes plants that are wild from the start, but omits species that have no seed sources within dispersal range or that cannot establish under current site conditions (Bauld et al., 2023). A review of tropical forest recovery after land abandonment found that, while some structural and functional properties recovered in 20–60 years, biomass and species composition took more than 120 years (Poorter et al., 2021). In Europe, recent woodlands (< 120 years old) developed by passive rewilding had higher taxonomic, functional and phylogenetic richness than older woodlands (>230 years), but very different species compositions (Morel et al., 2020). The absence of many species characteristic of older woodlands largely reflects dispersal limitation, but agricultural legacies, such as raised nutrient levels, may also exclude some species. Dispersal limitation is also a likely explanation for the absence of numerous regionally rare species from mesic grasslands on reclaimed marine sediments at Oostvaardersplassen 30 years after rewilding with large herbivores (Ejrnæs et al., 2024). At grassland sites in Germany, experimental seed additions with and without disturbances confirmed the importance of both dispersal and establishment limitations in excluding species (Freitag et al., 2021).

If more rapid recovery is required (to suppress fires or invasive species, to create habitat for animals, for aesthetic reasons or to provide other ecosystem services), natural regeneration can be encouraged by controlling competitors, prescribed burning, excluding or reintroducing herbivores or fertilisation. Initial planting can also be used to overcome dispersal barriers and compensate for land-use legacies, before allowing natural processes to dominate (e.g., Adair and Ashmole, 2024). In previously forested areas, trees may be planted over part of a site in order to attract dispersal agents and provide the shade and microclimate that many species need to establish ('applied nucleation'; Werden et al., 2022), or over the whole site with the aim of promoting subsequent diversification by natural dispersal and establishment (Elliott et al., 2022). In addition to providing shade and a favourable microclimate, the species planted may be chosen to favour those that are unlikely to reach the site without assistance due to source, dispersal or establishment

limitations. An experimental comparison in southern Costa Rica of natural regeneration, applied nucleation and planting the whole site found that active tree planting accelerated the establishment of late-successional species compared with natural regeneration, and planting the whole site increased the establishment of larger-seeded species (Schubert et al., 2025). Longer-term studies are needed to determine whether these early differences persist.

Similar considerations apply to rewilding monoculture plantations as to open sites, but the presence of an existing canopy creates additional choices. A quantitative review of 68 studies found that the diversity and abundance of native species established increased with proximity to native forest remnants (Kremer and Bauhus, 2020). Thinning the canopy and understorey also promoted colonisation in some studies. In an oil palm plantation in Sumatra, planting larger and more diverse tree islands increased recruitment diversity (Paterno et al., 2024).

## Soil issues

The rewilding literature has little to say about soils because they are rarely an obvious barrier to animal reintroductions. Plants, however, are far more dependent on soil conditions. Anthropogenic land-use change almost always causes changes in soil structure, chemistry and/or biology, and a third of the global land surface has changed land-use since 1960 (Winkler et al., 2021). Land-use soil legacies are both abiotic, including soil loss, compaction and nutrient excesses and/or deficiencies, and biotic, including changes to the seed bank, microbiota and soil fauna. After short-term cultivation, soil recovery can be rapid (Poorter et al., 2021). However, after prolonged cultivation, recovery is still incomplete after several decades, and full recovery is expected to take centuries or millennia (Parkhurst et al., 2022).

Physical manipulations, such as topsoil removal to reduce excess soil nutrients, fertilisation to replace missing nutrients, soil addition to replace material removed or lost to erosion and topographic modification are generally practical only in small areas, such as abandoned industrial and mining sites (König et al., 2022). However, while soil amendments and tree planting can restore vegetation structure, the restoration of plant diversity is still largely dependent on seed dispersal from nearby natural vegetation. Despite this, vegetation will eventually develop if the area is left alone, even on the most challenging brownfield sites (Trueman et al., 2022). Such vegetation may have no natural counterparts; however, it is wild and may be of conservation interest.

## The need for active management

The rewilding ideal is to restrict active management to the initial stages. Thereafter, with the system on a desired trajectory, it can be left to develop on its own, with continued intervention only necessary in small areas or where it is impractical to reintroduce keystone vertebrates. However, for plants and vegetation, this ideal is only likely to be achieved in small, little-degraded areas surrounded by intact native ecosystems. Almost everywhere else, agricultural legacies and dispersal limitation (Isbell et al., 2019) mean that some degree of active management, such as burning and re-seeding of restored prairies (McFarlane et al., 2023) and enrichment planting of restored forests (Sangsupan et al., 2018), will usually need to continue. The alternative, without management, may be wilder, but it will not be as diverse as it could be and may not fully restore ecosystem functions. There is also a risk of dominance by non-native invasive species.

## The roles of animals

It is almost an act of faith in trophic rewilding that the restoration of large herbivores will benefit vegetation recovery by suppressing competitive dominants, creating heterogeneity and dispersing seeds. Although the top-down control of plant communities by terrestrial herbivores is globally widespread, its strength is site-specific and the effects on plant diversity are not consistent (Jia et al., 2018). Plant communities and individual plant species are also strongly influenced by bottom-up factors, including soils and topography. There have been an increasing number of rewilding experiments designed to investigate the impacts of large herbivore introductions on grassland plants and arthropods, but it is not clear how far the usually positive results of these short-term studies can be generalised (e.g., Garrido et al., 2019; Bonavent et al., 2023).

A meta-analysis of the impacts of extant wild megaherbivores on ecosystems found that, in general, they promote open vegetation structure and spatial heterogeneity (Trepel et al., 2024). At the plot scale, communities dominated by these non-selective bulk feeders tend to have increased plant diversity in comparison to those dominated by smaller, more selective feeders (Lundgren et al., 2024). *A priori*, we would expect the strongest effects from the reintroduction of native megaherbivores in landscapes that still retain their native flora, and this is borne out by the impacts of reintroducing bison to tallgrass prairie in North America (Ratajczak et al., 2022). On the other hand, the current high densities of bison in Yellowstone National Park are contributing to the biotic impoverishment of riparian plant communities (Kauffman et al., 2023). Large domestic or feral mammalian herbivores may, at least partly, substitute for missing native species, and a global meta-analysis showed that herbivore functional traits were more important than nativeness in determining their effect on plant communities (Lundgren et al., 2024).

The reintroduction of extirpated seed dispersal agents to enhance the number and diversity of seeds dispersed into a site has also been widely advocated. Modelling studies support the effectiveness of this approach, but empirical evidence is still limited (Mittelman et al., 2022). Where dispersal agents are present in the landscape, artificial perches may increase the density and diversity of the seed rain (Mayta et al., 2024).

Moving up a trophic level, it has been claimed that reintroducing large carnivores can help restore plant communities through a trophic cascade, mediated by their influence on herbivore numbers and behaviour, but evidence for this is mixed (Clark-Wolf and Hebblewhite, 2024). At Yellowstone, for example, the restoration of large carnivores after almost a century of absence has failed to restore riparian plant communities in the northern range, suggesting a possible alternative stable state (Hobbs et al., 2024).

The top-down benefits of rewilding with large vertebrates are even less clear for individual plant species. Although the impacts of large herbivores on plants are typically painted in broad-brush terms in the rewilding literature—feeding, seed dispersal and trampling; wallowing and other disturbances—in reality, they differ greatly among plant species. As a result, the rewilding of rare plant species is as likely to require the exclusion of herbivores as their reintroduction (Silcock et al., 2019; Adair and Ashmole, 2024).

These examples warn against relying solely on large vertebrate reintroductions to restore rare plant species and degraded plant communities. This may be largely a question of scale. In pre-human landscapes, the megafauna, mesofauna and other animals interacted with geodiversity (topography, soils, drainage etc.) to produce environmental heterogeneity at the landscape and local levels that provided sites in which populations of all native plant species could

persist. Contemporary rewilding initiatives cannot replicate this because of area limitations, extinctions, climate change and other human impact legacies.

## Can we rewild plants and vegetation?

Rewilding plants has many challenges, including persistent land-use legacies, dispersal limitation, invasive alien species and the uncertain impacts of animal rewilding. In comparison with large vertebrates, however, there have been relatively few known plant extinctions and most species still have wild populations. Moreover, natural processes will cover almost any abandoned site with wild vegetation within months or years. The biggest uncertainty is the future of the numerous narrow-range specialists, which individually play minor roles, but collectively account for most plant diversity and almost all endangered plant species, and significantly contribute to the diversity of ecological functions performed by plants (Mouillot et al., 2013). Where protecting and re-establishing these species is a goal of rewilding, additional human intervention will almost always be needed (e.g., Ejrnæs et al., 2024).

Taxon substitutions, such as domestic or de-domesticated grazers for extinct herbivores, can make sense for functionally important vertebrates (Lundgren et al., 2024), but there is rarely a similar justification for plants. Possible exceptions are some island tree species that became extinct following human settlement, including a palm species that dominated forests on Easter Island and a *Quercus* on Tenerife. Conservation introductions outside the native range (assisted colonisations) are controversial, but likely to be increasingly necessary when the original habitat is no longer suitable, because of climate change or other irreversible impacts (Christenhusz and Govaerts, 2025). In many cases also, plants are wanted for their structural and functional roles more than for their specific identities, and the aim of including them in rewilding practice is to accelerate the recovery of a natural (or naturalistic) habitat structure for the benefit of both plants and animals ('foundation plants'; Root-Bernstein et al., 2024). When plants are viewed in this way, careful taxon substitutions and conservation introductions are more easily justified.

Our uncertain ability to successfully rewild plant species emphasises the need to prevent these losses in the first place. In most cases, protecting an existing population will be easier and far less expensive than attempting to re-establish it after it has gone. Where *in situ* conservation is not possible, a range of *ex situ* options means that almost all plant species can be saved from extinction (Corlett, 2023). If and how these species can eventually be returned to the wild is unclear, however, and some will likely need continued care, either in captivity or in a semi-wild state.

## Conclusions and recommendations

Passive rewilding of plant species and vegetation occurs spontaneously whenever human pressures are released, regardless of the state of the site. Human intervention is necessary only if recovery is slower than desired or not on a desired trajectory. The diversity of native plant species that appear will depend on site conditions, the proximity of seed sources and the availability of dispersal agents. In most cases, the full recovery of native plant diversity will require human intervention, through initial planting or sowing and/or later enrichment with species that do not arrive naturally. The reintroduction of dispersal agents may also be useful. For plant conservation, the key issue is ensuring that all plant species that can be restored are present, including rare and threatened species. Long-term species-level monitoring and, where necessary, continued intervention should be part of all projects that aim to rewild plants and vegetation.

**Open peer review.** To view the open peer review materials for this article, please visit http://doi.org/10.1017/ext.2025.1.

**Data availability statement.** The author confirms that the data supporting the findings of this study are available within the article.

**Acknowledgements.** The author is grateful to the handling editor and two anonymous reviewers for their useful comments and suggestions.

**Financial support.** This research received no specific grant from any funding agency, commercial or not-for-profit sectors.

**Competing interest.** The authors declare none.

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
