## [Reviewer Report]

This is an interesting and important perspective.

While I appreciate the focus, it would be helpful to see more explicit acknowledgement and clarification of ecological restoration vs rewilding. Additionally, in response to sections about the establishment of plant populations- more explicit acknowledgement and points of conservation biology. An acknowledgement of where the venn diagram of these fields overlap and differ in terms of a plant world view would be helpful, perhaps even a visual venn diagram.

I also think there should be more careful attention to citing facts, and to citations in general. I’d like to see more appropriate citations for restoration and conservation biology.

While rewilding is interesting, the trouble with it, is the difficulty in measuring it, and thus the lack of data to support the ideas. In my opinion, this makes it a lot of well-sounding talk, without much evidence to support actions. An acknowledgement of this difficulty and these limitations, in contrast to ecological restoration which is increasingly able to take a more science-driven and evidence-based approach would be helpful, especially to new readers on the topic. This is largely washed over by the rewilding field, which has disappointment.

---

## [Reviewer Report]

Thank you for the invitation to review the manuscript “Rewilding plants and vegetation” by Richard T. Corlett. I thoroughly enjoyed reading this well-written perspective piece and enjoyed the unique view it brought to the field. In particular, explicitly linking and highlighting the potential asymmetry between the attention given to top-down processes in trophic rewilding versus plants and vegetation, then outlining the potential pitfalls of doing so.

I do not agree with all the sentiments within the paper. However, this is indeed a perspective piece, so my main comments are more about specific parts of the literature that may be worth including that haven’t been. Overall, I think this piece will promote valuable and interesting discussion in a burgeoning field of applied ecology.

Broad comments:

For instance, that it is an “act of faith” that rewilding will benefit vegetation via a range of ecological processes. I offer at the bottom of this review some experimental evidence from the rewilding literature and broader plant-herbivore interaction literature that may be incorporated to provide some relevant context for this point. And indeed, the author cites some relevant studies that show support for at least some of these processes on a large scale (Lundgren et al. 2024; Trepel et al. 2024). While it is true that much of the rewilding enthusiasm has yet to present strong peer-reviewed evidence for many aspects of vegetation recovery, the mechanistic logic is not without scientific support. The author references Jia et al. 2018, stating the evidence is mixed, though from a land manager point of view such syntheses are often made up of positive and negative effects. However, viewed through a trait lens, considering body size of the herbivores is often found to be particularly important (e.g. in Jia et al. 2018, as well as Trepel et al. 2024 though this is more about effects shifting from negative to neutral). Working on resolving such idiosyncrasies remains a timely objective for rewilding scientists.

One other broad comment is it may be relevant to highlight that the lack of consideration of dispersal dynamics and rare species is not unique to rewilding – it is also an issue in vegetation-focused restoration activities. Is this bias or gap particularly pronounced in rewilding than in other land management practices do you think?

Perhaps relevant would be some of the literature around “applied nucleation”, which focuses directly on dispersal improvement in degraded landscapes and thus may be considered as a plant-centric form of rewilding (in terms of a process-focused method).

Relevant literature regarding more manufactured disturbance such as Freitag et al. 2021 while not a test of rewilding, also inform the scientific logic around incorporating seeding alongside disturbance processes - just a thought.

A relevant recent study that does investigate the missing diversity of rare species at the Oostvaardersplassen rewilding site in the Netherlands (Ejrnæs, et al. 2024) while not incorporating strategies to promote them is starting to address this and might be relevant to such discussion.

Minor comments:

L47 A relevant reference perhaps for this point on megabiota, Enquist et al. 2020

L56-58 A point here may be that while the goal of rewilding is aimed to be minimized, it is in fact massively taxing and labor intensive under many settings to work with large animals, due to ethics concerns etc, so while this is the goal, it is not always true

L61-62 Perhaps at the core of rewilding, and a bit tangential to this piece, but what is “wilderness” as a goal? Many of the biodiverse landscapes rewilding aims at restoring, e.g. open grassy forb-rich systems, while “wild”, have persisted and been promoted in many settings as a result of human land use (which incidentally is part of their image problem), e.g. in Eastern Europe by pastoralists.

L128 I will admit at this stage I got a bit lost in the terminology of rewilding versus restoration. Natural regeneration of plants traditionally would fall under the remit of restoration – passive restoration – or even just might be considered vegetation recovery. Not sure I see the need to refer to natural regeneration as passive rewilding, but that is a minor point.

L193 A major argument from the plant side, which you make elsewhere but might be relevant to this point, is not just dominance by non-natives, but just dominated by a few well dispersed species, with dispersal and fragmentation limiting natural recruitment or “volunteer species”, at least on human lifetime timescales.

L195 Perhaps especially, or more so, with those taxa that have relatively strongly related living congeners? E.g. Equus in North America in the paper cited

L211-213 While this is true there are many exceptions to this rule as you say above, and it is largely an emergent statistical property. So, the practical challenge remains, what do you do with large herbivores at the sites similar to those included in the study that show conflicting results?

213-215 The logic here is not clear to me. Why? And strongest effects on what components of the ecosystem?

L252 “most species are still wild somewhere” is a phrase repeated from earlier (L85), suggest rewording

References

Enquist, B. J., Abraham, A. J., Harfoot, M. B. J., Malhi, Y., & Doughty, C. E. (2020). The megabiota are disproportionately important for biosphere functioning. Nature Communications, 11(1), Article 1. https://doi.org/10.1038/s41467-020-14369-y

Ejrnæs, D. D., Olivier, B., Bakker, E. S., Cornelissen, P., Ejrnæs, R., Smit, C., & Svenning, J.-C. (2024). Vegetation dynamics following three decades of trophic rewilding in the mesic grasslands of Oostvaardersplassen. Applied Vegetation Science, 27(3), e12805. https://doi.org/10.1111/avsc.12805

Freitag, M., Klaus, V. H., Bolliger, R., Hamer, U., Kleinebecker, T., Prati, D., Schäfer, D., & Hölzel, N. (2021). Restoration of plant diversity in permanent grassland by seeding: Assessing the limiting factors along land-use gradients. Journal of Applied Ecology, 58(8), 1681–1692. https://doi.org/10.1111/1365-2664.13883

Some additional references for the positive effects of rewilding (though all from Europe)

Bonavent, C., Olsen, K., Ejrnæs, R., Fløjgaard, C., Hansen, M. D. D., Normand, S., Svenning, J.-C., & Bruun, H. H. (2023). Grazing by semi-feral cattle and horses supports plant species richness and uniqueness in grasslands. Applied Vegetation Science, 26(1), e12718. https://doi.org/10.1111/avsc.12718

Garrido, P., Mårell, A., Öckinger, E., Skarin, A., Jansson, A., & Thulin, C.-G. (2019). Experimental rewilding enhances grassland functional composition and pollinator habitat use. In Journal of Applied Ecology (Vol. 56, Issue 4, pp. 946–955). https://doi.org/doi.org/10.1111/1365-2664.13338

Klink, R. van, Laar-Wiersma, J. van, Vorst, O., & Smit, C. (2020). Rewilding with large herbivores: Positive direct and delayed effects of carrion on plant and arthropod communities. PLOS ONE, 15(1), e0226946. https://doi.org/10.1371/journal.pone.0226946

Mutillod, C., Buisson, E., Tatin, L., Mahy, G., Dufrêne, M., Mesléard, F., & Dutoit, T. (2024). Managed as wild, horses influence grassland vegetation differently than domestic herds. Biological Conservation, 290, 110469. https://doi.org/10.1016/j.biocon.2024.110469

---

## [Editor Report]

Both reviewers appreciate the aim of this perspective and consider that it will spark significant and constructive discussions on the topic.

The author should consider disambiguating some statements, referencing additional literature, and exploring the overlaps and differences in rewilding vs. ecological restoration.

---

## [Reviewer Report]

The author has not attempted to respond with changes to their work for any one of my comments, and thus, I don’t feel particularly compelled to spend time offering more suggestions. I didn’t suggest a review of restoration or getting involved in the debate, but rather, clarifications. Attention to citations is still lacking. For example, lines 52-60 lack citations. Line 60-65 is pretty long for a sentence. I still really appreciate this work, but I get the impression it has not been carefully proofread or that feedback is not genuinely sought and considered.

---

## [Reviewer Report]

Thank you for the opportunity to see this manuscript for a second time. The author has clearly responded to my previous comments, justifying not changing the text or adding where they deemed appropriate. As I said in my first review, it was not my intention to insert my own opinion into this perspective piece, and therefore I would be happy to see this perspective piece accepted for publication.

---

## [Editor Report]

This revised submission addressed many of the reviewer comments, within the context and constraints of a perspective piece on rewilding plants that is also part of a special issue. Enough key statements were reworded to avoid ambiguity and clear up the conceptual differences with ecological restoration. Missing references and literature pointed out by one of the reviewers were also added. I am satisfied with the edits and ultimately with the discussion and debate that the paper will spark and fit into.

---

## [Reviewer Report]

This work effectively presents the unique considerations of vegetation in rewilding contexts. Nicely done!

---

## [Editor Report]

Both reviewers are satisfied with the changes made to address the minor comments from the previous version and I agree with their assessment. Accept.